# Novel Effective Photocatalytic Self-Cleaning Coatings: TiO_2_-Polyfluoroalkoxy Coatings Prepared by Suspension Plasma Spraying

**DOI:** 10.3390/nano13243123

**Published:** 2023-12-12

**Authors:** Chunyan He, Jialin He, Sainan Cui, Xiujuan Fan, Shuanjian Li, Yaqi Yang, Xi Tan, Xiaofeng Zhang, Jie Mao, Liuyan Zhang, Changguang Deng

**Affiliations:** 1School of Materials and Energy, Guangdong University of Technology, Guangzhou 510006, China; 2The Key Lab of Guangdong for Modern Surface Engineering Technology, National Engineering Laboratory for Modern Materials Surface Engineering Technology, Guangdong Institute of New Materials, Guangdong Academy of Sciences, Guangzhou 510651, China; 3College of Mechanical and Electrical Engineering, Shaanxi University of Science and Technology, Xi’an 710021, China; 4Qingdao Haier Refrigerator Co., Ltd., Qingdao 266510, China

**Keywords:** hydrophobic, photocatalytic, suspension plasma spraying, TiO_2_-PFA

## Abstract

Photocatalytic coatings can degrade volatile organic compounds into non-toxic products, which has drawn the attention of scholars around the world. However, the pollution of dust on the coating adversely affects the photocatalytic efficiency and service life of the coating. Here, a series of TiO_2_-polyfluoroalkoxy (PFA) coatings with different contents of PFA were fabricated by suspension plasma spraying technology. The results demonstrate that the hybrid coatings contain a large number of circular and ellipsoidal nanoparticles and a porous micron-nano structure due to the inclusion of PFA. According to the optimized thermal spraying process parameters, TiO_2_ nanoparticles were partially melted to retain most of the anatase phases, whereas PFA did not undergo significant carbonization. As compared to the TiO_2_ coating, the static contact angle of the composite coating doped with 25 wt.% PFA increased from 28.2° to 134.1°. In addition, PFA strongly adsorbs methylene blue, resulting in a greater involvement of methylene blue molecules in the catalyst, where the catalytic rate of hybrid coatings is up to 95%. The presented nanocomposite coatings possess excellent photocatalytic and self-cleaning properties and are expected to find wider practical applications in the field of photocatalysis.

## 1. Introduction

Photocatalytic surfaces have become increasingly popular in recent years, since their discovery in 1969 by Fujishima and Honda [1]. TiO_2_ coatings have attracted much attention in recent years due to their potential for photocatalysis [2]. However, traditional TiO_2_-based photocatalytic surfaces are superhydrophilic, with low light utilization and easy dust adsorption, both of which are bad for photocatalysis. Superhydrophobic surfaces can easily roll over and take away the surface pollutants, thereby maintaining the cleanliness of the photocatalytic coating, ensuring its mechanical strength, and preserving its photocatalytic capabilities [3,4].

While most studies have focused on the TiO_2_ superhydrophilic photocatalytic surfaces, there has been little research on TiO_2_ superhydrophobic photocatalytic surfaces. Typically, TiO_2_-based photocatalytic superhydrophobic coatings are produced using chemical methods, such as hydrothermal [5,6], liquid phase deposition [7], sol–gel [8], etc. The hydrothermal modification of TiO_2_ with silicone rubber results in self-cleaning surfaces with a multilevel structure that is highly hydrophobic and photocatalytic [9]. Sanghyuk et al. [10] created a self-cleaning surface with both superhydrophobicity and photocatalytic activity by grafting PDMS molecular brushes with TiO_2_. Several studies have used photocatalytic materials other than TiO_2_, such as SiO_2_, for the preparation of superhydrophobic surfaces. By using the sol–gel method, Tian et al. [11] developed PDMS–TiO_2_–SiO_2_ coatings with hydrophobic photocatalytic properties. Using dip-coating, Jiang et al. [12] prepared fluorine-free superhydrophobic photocatalytic synergistic self-cleaning functional cotton fabrics by modifying cotton fabric surfaces with anatase TiO_2_ and PDMS. In spite of the fact that all of the above processes prepared TiO_2_-based hydrophobic photocatalytic coatings, there were issues, such as many preparation steps, complicated processes, uneven coating formation, and lower photo-catalytic properties. It is a rather challenging task combining superhydrophobic with considerable photocatalytic activity in TiO_2_-based coatings, as the TiO_2_ coating typically exhibits hydrophilic properties, turning superhydrophilic upon ultraviolet (UV) light exposure.

Superhydrophobicity is typically described as resulting from nano- and micro-scale roughness in combination with the hydrophobic functionality of the material [13]. This study used the simple, easy, low-cost, fast, and efficient suspension plasma spraying (SPS) method [14,15]. SPS can better construct a micro-nano porous porous coating by solvent evaporation, which is advantageous to photocatalytic performance. Meanwhile, a typical low surface energy substance, polytetrafluoroethylene PFA was selected [16]. The low surface energy of PFA provides the necessary conditions for hydrophobicity. In detail, a TiO_2_-PFA hydrophobic photocatalytic coating was constructed in one step through the spraying of a suspension of TiO_2_ nanopowder and PFA emulsion using SPS. It is possible to prepare hydrophobic photocatalysis TiO_2_/PFA coatings over a large area quickly and efficiently. Because SPS uses the evaporation of the solvent to remove a large amount of heat, and the droplets are ejected from the flame flow at very high speeds, the flight time of droplets is very short, and the heat absorbed by the solid particles in the droplets is sharply reduced [17,18,19,20], resulting in fewer anatase transformations to the rutile phase in the suspension plasma sprayed TiO_2_ particles in this study. In the meanwhile, PFA is not decomposed by carbonization. This ensures the photocatalytic and hydrophobic properties of the PFA/TiO_2_ coating.

In the present work, the TiO_2_-PFA hydrophobic photocatalytic self-cleaning coatings were prepared in a one-step process using suspension plasma spraying. The self-cleaning and photocatalytic properties of the developed TiO_2_-PFA composited coatings against contaminants were investigated under UV-light irradiation. This study’s main aim was to evaluate the suitability of the suspension plasma spraying method for superhydrophobic photocatalytic coatings and the potential applications of TiO_2_-PFA coating in outdoor contamination prevention.

## 2. Experiment Methods

### 2.1. TiO_2_-PFA Suspension

In this suspension, anatase TiO_2_ powder with an average particle size of 200 nm was combined with polyfluoroalkoxy (PFA, 60 wt.%) emulsion, dispersant NNO, and deionized water. The mass of PFA accounted for 0%, 5%, 15%, and 25% of the total mass of TiO_2_-PFA, and the corresponding suspensions and prepared coatings were named as: 0PT, 5PT, 15PT, and 25PT, respectively. The specific components of each suspension are shown in Table 1.

### 2.2. Coatings Preparation

(1)A stainless-steel sheet with dimensions of 20 mm × 20 mm× 2 mm was employed as the substrate and sandblasted using 46# SiO_2_. It was then cleaned ultrasonically with alcohol and dried using compressed air.(2)For suspension plasma spraying, an F4 plasma gun (GTV CL, WI-091) with an inner diameter of 6 mm was employed. The suspension was dispersed and stirred in real-time using a custom-made thermostatic magnetic stirring and ultrasonic vibration machine, ensuring good fluidity and stability of the suspension during the SPS spraying process (Figure 1). A peristaltic pump was utilized to convey the TiO_2_-PFA suspension from a 0.3 mm diameter injection port into the plasma stream at a rate of 30 mL/min. The spraying parameters are shown in Table 2. Importantly, cooling the substrate with air during the spraying process can be beneficial for the retention of anatase TiO_2_ and avoiding the over-firing carbonization decomposition of PFA.

### 2.3. Coating Characterization

Additional insights into the surface morphology were acquired via a field emission scanning electron microscope (FE-SEM; Hitachi, SU8010, Tokyo, Japan). The chemical composition of the coatings was determined using a Philips X-ray diffractometer (XRD, X’Pert MPD, Royal Philips, Amsterdam, The Netherlands; Cu-Kα radiation, potential 40 kV, current 30 mA), Fourier transform infrared spectroscopy ((ATR-FTIR, Nicolet IS5), and X-ray photoelectron spectroscopy (XPS, ThermoSCIENTIFICESCALAB250Xi, Waltham, WA, America). Static water contact angles (WCA) were measured with a Theta Lite optical tensiometer. The average contact angle values were obtained by taking a mean value of 5 measurements across the sample surface.

### 2.4. Evaluation of Photocatalytic Activity

The UV–Vis analysis was performed using a UV–Vis diffuse reflectometer (UH4150, Hitachi, Japan) to examine the light absorption capacity of the prepared samples in the wavelength range of 200~800 nm. The band gap value of TiO_2_ was determined by generating a Tauc plot curve from the UV–Vis diffuse reflectance data. The absorption coefficient α is related to the bandgap Eg in accordance with Equation (1):(1)$${\left(\alpha \mathrm{hv}\right)}^{1/n}=z\;\times \;(\mathrm{hv}\;-\;\mathrm{eg})$$
where n = 0.5 for direct bandgap semiconductors, n = 2 for indirect bandgap semiconductors, hν is the incident light energy hν = 1240/λ, λ is the incident wavelength, and Z is a constant [21].

A photochemical reactor (Guangzhou Xingchuang Electronics Co., Ltd., Guangzhou, China, XC-BU921-2) was used as the photocatalytic reaction device, with a light intensity of 2 mW/cm^2^ for UV light and 30 mW/cm^2^ for visible light, and its UV portion was shielded using a cut-off filter of 420 nm. Methylene blue (MB, Beilian Fine Chemicals Co., Ltd. Tianjin, China) solution was used to simulate polluted water, and the absorbance of MB solution was tested by a spectrophotometer (Spectrum 752PC, Shanghai, China) as a means to evaluate the photocatalytic activity of the coatings [22]. The dried coating samples were placed in a jacketed beaker containing circulating water, and the MB solution (30 mL, 5 ppm) was added to completely immerse the samples. The circulating cooling water was implemented to mitigate the impact of solution evaporation on the results during light exposure. First, the entire device was kept in darkness for one hour, then an appropriate amount of MB solution was removed, and the concentration of MB solution was recorded by utilizing a UV–Vis spectrophotometer to detect the absorption value of the solution at 664 nm. This concentration was determined as the concentration C0 after the dark reaction. The sample was introduced into the reactor while the UV/visible light lamp was activated. An appropriate amount of MB solution was withdrawn every 20 min, and its concentration was recorded as Ct. After the test, the solution was returned to the beaker. The residual rate P of the MB solution was calculated by Equation (2).
(2)$$P=\frac{C_{t}}{C_{0}}\times 100\%$$
where C_0_ is the concentration of the solution after the dark reaction, and C_t_ is the concentration of the MB solution at time t.

By using kinetic modeling, the degradation results of coatings and powders were compared more intuitively. Due to the fact that, at a certain concentration, the concentration of organic dyes changes with time in accordance with the Langmuir–Hinshelwood first-order kinetic equation [23], namely:(3)$$\left(-{\mathrm{dC}}_{t}\right)/\mathrm{dt}={\mathrm{kC}}_{t}.$$

Calculating Formula (3) using both sides yields:(4)$$-\ln\left(\frac{C_{t}}{C_{0}}\right)=\mathrm{kt}\;$$
where k indicates the slope of the straight-line fitting illumination time (t) and ln (C_0_/C) are the horizontal and vertical coordinates, which represent the apparent rate constants.

## 3. Results and Discussion

### 3.1. Coating Microstructure 

Figure 2 illustrates the corresponding surface morphologies of 0PT, 5PT, 15PT, and 25PT. It was observed that the coating was all composed of micro- and nano-sized clusters of small circular or ellipsoidal particles—the formation of the melt semi-melt agglomerations when TiO_2_ nanopowder particles are heated by flame flow. The solvent evaporates instantaneously, depositing onto the substrate and creating numerous pores within the coating. The surface of the composite coatings associating TiO_2_ and PFA exhibited a flocculent drawing substance, with the mass of this flocculated, drawn substance increasing as the PFA content increased.

As the amount of PFA content approached 25%, the flocculated drawing material took on a net-like structure, from an irregular, one-dimensional flocculation to a two-dimensional, net-like structure. Table 3 demonstrates the average surface roughness of the coatings was influenced by surface morphology of the coatings. Presumably, as a result of the flocculent material produced, it is evident that the surface roughness of the coating increases as the PFA content rises. The hydrophobic nature of PFA results from the low surface energy and the surface roughness of microporous surfaces.

To further investigate the distribution state of the PFA in the coatings, surface scanning analyses were performed on the coatings, as depicted in Figure 3. As shown, the elemental distribution of Ti and F was extremely homogeneous, and as the PFA concentration rose, the prominence of the green hue on the coating’s surface increased. This correspondence signifies the presence of a higher concentration of F elements in the coating. The hydrophobic nature of the coating is attributed to the presence of the F element, which is derived from the low surface energy component of the PFA. The Ti element serves as a photocatalyst within the coating, and the content of the PFA does not appear to impact the amount of TiO_2_ on the surface. Consequently, the coating is capable of reacting with oxygen or water in the air under certain wavelengths of light, leading to the generation of redox active molecules.

The cross-sectional structure and EDS analysis of different coatings are presented in Figure 4. The findings reveal a striking uniformity in the distribution of the F element content across the composite coatings. Different from the irregular distribution of the F element in the Al_2_O_3_/PFA wear-resistant hydrophobic coating prepared directly by atmospheric plasma spraying [24], the ceramic–polymer composite coating was prepared by suspension plasma spraying in this study, so that the F element was evenly distributed in the composite coating, thus, ensuring the stability of the coating performance and excellent hydrophobicity.

### 3.2. Coating Composition

The XRD patterns of the TiO_2_, PFA powder (PFA emulsion dried at 150 °C), and coating are shown in Figure 5a,b, respectively. The PFA powder had the main crystalline phase (C_2_F_4_)n, whereas the diffraction peaks of TiO_2_ were 25.3°, 53.9°, 68.8°, 70.3°, and 78.7°, which corresponded to the (101), (105), (116), (220), and (206) crystal planes of the anatase phase of TiO_2_, respectively [25]. The diffraction peaks at 41.2° and 62.7° corresponded to the (111) and (002) crystal faces of the TiO_2_ rutile phase, respectively [26]. As shown in Figure 5b, the coating obtained by SPS had different peaks than the original powder. The plasma spraying of the TiO_2_ coatings results in mostly anatase TiO_2_, but also rutile and TiO_2_-x. The phase composition of composite coatings is essentially the same, but their anatase content differs. The coatings exhibited the presence of both anatase and rutile phases of TiO_2_; some additional crystal planes also started to appear, such as the (113) crystal planes of the TiO_2_ anatase phase and the (110) crystal planes of the TiO_2_ rutile phase. The presence of this phenomenon verifies the existence of TiO_2_ anatase–rutile heterojunctions in the coatings, and the heterojunctions still existed after the introduction of PFA. The formation of this heterojunction suppresses the recombination of photogenerated carriers to a certain extent, thus improving the photocatalytic activity [27,28]. Table 4 shows the anatase content of the coatings. The amount of anatase in the 5PT coating is comparable to that in the 0PT coating. As the amount of PFA content increased, the concentration of anatase in the coating grew accordingly. The contents of the PFA exhibited a moderate alleviating effect on the conversion of anatase to rutile. It may be due to the fact that the PFA in the coating consumes a fraction of the heat in the flame stream, thereby exposing the TiO_2_ powder particles to less heat and preserving the anatase phase. As the PFA content and anatase content in the coating increase, the coating will exhibit hydrophobic catalytic properties.

### 3.3. FT-IR Analysis of Composite Coatings

In general, APS technology is commonly employed for the preparation of metal or metal oxide coatings, while high molecular weight polymers are not suitable for APS due to their characteristics, such as high-temperature resistance and thermal decomposition. Therefore, it is necessary to investigate whether PFA will undergo decomposition after SPS technology. Gawne et al. successfully prepared polymer coatings using APS technology, and by optimizing parameters related to spraying, they were able to melt the white powders of the polymer without decomposition or defluorination [29]. As illustrated in Figure 6, infrared spectroscopic tests were conducted on PFA powder (produced by drying a PFA emulsion at 150 °C) and composite coatings, respectively, to better understand the form of PFA present in the coatings. The results indicated that the peaks in the composite coatings (1000–1400 cm^−1^) remained consistent with those observed in the initial PFA powder, with minimal displacement variation in the distinctive peaks. The polytetrafluoroethylene target spectrum had two characteristic peaks at 1201 cm^−1^ and 1150 cm^−1^ that were attributed to asymmetrical and symmetrical CF_2_ stretching. A third weaker peak corresponding to the CF_2_ wagging was observed at 642 cm^−1^, which is consistent with the literature data [30]. This meant that the PFA in the coatings was about the same as it was before spraying. Research findings indicate that the SPS process melts PFA without inducing decomposition or stripping, hence preserving the hydrophobic characteristics of the coating.

### 3.4. Composite Coating XPS Analysis

To further study the chemical composition of the surface the cotings, the element contents of PFA powder and coating were determined by using XPS. Figure 7 shows the XPS spectra of PFA powder. As a result of the spectral analysis, it is evident that the characteristic peaks of PFA powder are the F, C, and O peaks. Figure 8 displays the XPS spectra of the composite coatings. Obviously, the surface elements of composite coatings are primarily composed of C, O, F, Si, and Ti, respectively. Figure 8d shows the Ti 2p spectrum composed of two dominant peaks positioned at 457.68 and 463.38 eV, which specifies the characteristic peak of Ti 2p_3/2_ and Ti 2p_1/2_, respectively [31]. On the surfaces of the composite coatings, the F peak is always the main peak, and the Ti and O peaks are significantly weaker than the F peak in intensity. The vast C-F bonds with low surface energy from the PTFE powder are enriched onto the surface of the as-prepared coatings. This can be directly interpreted that the C-F bonds on the surface of coatings are constituted as the “material-level” factor for forming a superhydrophobic surface [32]. These results reveal the presence of a low surface energy material, fluoropolymer (PFA) on the coating’s surface, which contributes to its hydrophobic properties.

### 3.5. Effect of PFA Content on Light Absorption Performance of Coatings

Figure 9 depicts the results of the coating light absorption performance test. Figure 9a shows the UV–visible absorption spectrum of the sample. As can be seen from Figure 9a, compared with 0PT, with the increase of PFA content in the composite coating, the UV absorption intensity of the coating increased, while the visible light absorption intensity decreased; it may be caused by the PFA being deposited on the coating surface, reducing the effective light absorption area of the coatings.

The direct and indirect bandgaps of the coatings derived from extrapolating the straight-line portion of the (αhν)^2^-hν and (αhν)^1/2^-hν plots are shown in Figure 9b,c and Table 5. According to the table, both indirect and direct bandgaps of the composite coatings are larger than those of the 0PT. Nevertheless, in relation to the theoretical band gap value of TiO_2_ (3.2 eV), the coating exhibits a marginal reduction in the bandgap. It is speculated that during the synthesis of TiO_2_ coating by spraying, TiO_2_ undergoes reduction by H_2_ at elevated temperatures to facilitate self-doping modification with Ti^3+^, thereby significantly reducing the bandgap and enhancing the catalyst’s photocatalytic activity under visible light. The direct bandgaps of the composite coatings increased with increasing content, but the indirect band gap had little difference. The optical property of a semiconductor is considered one of the parameters affecting photocatalytic performance since its activity in the visible region can be enhanced by extending the light absorption into this region. In the semiconductor industry, any absorption edge shift is attributed to a change in the bandgap [33], and these results indicate that oxygen vacancies could effectively decrease the titania bandgap. An electron may be excited from the filling valence band of the band gap to the empty conduction band by the energy of the photon if the energy of the photon exceeds the bandgap. Among the two types of electron transitions from the valence band to the conduction band, indirect bandgap is mostly found in anatase, whereas direct bandgap is considered for rutile [34]. While, in the direct bandgap, electrons are ejected directly from the valance band maximum to the conduction band minimum; a momentum change is needed for the indirect bandgap transition, resulting in an increment of the charge carrier lifetime in anatase compared to that of rutile. This longer lifetime makes anatase more efficient for photocatalytic applications, as a larger number of the photo-generated electron-holes pairs are available to participate in surface reactions [35,36]. It is clear from the dark gray color of the 0PT that visible light is strongly absorbed. Normally, photocatalytic activity begins with the excitation of electrons by selective absorption. In addition, considering the different energy gaps of the rutile phase (Eg = 3.0 eV) and anatase phase (Eg = 3.2 eV), the main phase (anatase) has a different bandgap. It has been shown that the presence of the main phase (anatase) significantly affects the light absorption performance of the coatings, resulting in the presence of the main phase (anatase) significantly affects the light absorption properties of the sprayed coatings, which may lead to a higher indirect bandgap of coatings with an anatase-rich phase.

### 3.6. Effect of PFA Content on the Photocatalytic Performance of Coatings

The photocatalytic performance of the four groups of samples was examined by degradation of methylene blue (MB) under UV–Vis irradiation; the results are shown in Figure 10. In Table 6, the values of MB degradation after 360 min, the reaction rate constant (k), and the fitted correlation coefficient (R^2^) indicate the fitted first-order regression equation. Compared with the 0PT, the composite coatings exhibit higher photodegradation activity. As the content of PFA increases, the photocatalytic activity of 5PT, 15PT, and 25PT in visible light significantly improves, which can be attributed to the strong adsorption of PFA in the coating onto MB. As illustrated in Figure 10a,b, dark adsorption of the coating occurs between −60 and 0 in the horizontal coordinate, and 5PT, 15PT, and 25PT are more effective at adsorbing MB than 0PT. Due to the fact that the composite coatings are more likely to adsorb MB, there are more MB molecules involved in photocatalysis when the light starts at 5–6, so the catalytic rate of 5PT, 15PT, and 25PT is higher than that of 0PT. It is consistent with the view that substrate concentration plays a significant role in the catalytic process.

### 3.7. Effect of PFA Content on the Wettability of the Coating Surface

A material is considered hydrophobic when the hydrostatic contact angle exceeds 90°. Each composite coating displays excellent hydrophobicity, as seen in Figure 11. Figure 12 illustrates the contact angle value of the coatings. The hydrophobicity angles of the 5PT, 15PT, and 25PT were 110.5°, 123.5°, and 134.1°, respectively, all of which were larger than 90° and conferred hydrophobicity. In contrast, the 0PT included only TiO_2_ and had a contact angle of 28.2°. As the content of PFA increases, there is a gradual increase in the contact angle. The magnitude of the contact angle is influenced by two main parameters: surface roughness and surface energy controlled by surface chemistry [37,38]. According to the roughness results, the roughness increases as the PFA content increases, as does the contact angle. Furthermore, when the coating contains the low surface energy substance PFA, the contact angle of coating increases and its hydrophobicity is improved.

## 4. Conclusions

A novel hydrophobic photocatalytic self-cleaning composite coating was designed by combining photoactive TiO_2_ and the hydrophobic PFA. By using suspension plasma spraying, the effective photocatalytic self-cleaning TiO_2_-PFA coatings were successfully prepared on large surfaces without the complexity of using chemicals. The TiO_2_-PFA coatings present many circular and ellipsoidal nanoparticles with a flocculent porous micron-nano structure due to the PFA. The phase composition of TiO_2_-PFA coatings is mostly anatase TiO_2_, but also rutile and (C_2_F_4_)_n_ of PFA, as confirmed by XRD and XPS. Thus, the composite coating (PFA~25%) had a static contact angle of around 134°, indicating excellent hydrophobicity. MB, an organic contaminant adsorbed on the surface of the composite coating, was also successfully removed by photocatalytic oxidation by 95%, resulting in a sustained high hydrophobicity. These results demonstrate the potential application of this functional coating PFA/TiO_2_ in the development of photoactive hydrophobic materials for protecting against outdoor pollution.

## Figures and Tables

**Figure 1 nanomaterials-13-03123-f001:**
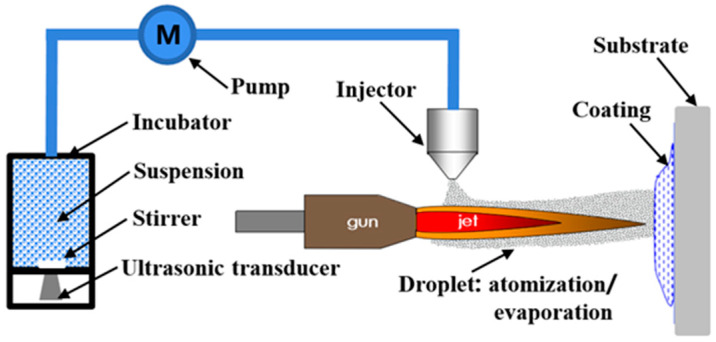
Schematic diagram of suspension plasma spraying.

**Figure 2 nanomaterials-13-03123-f002:**
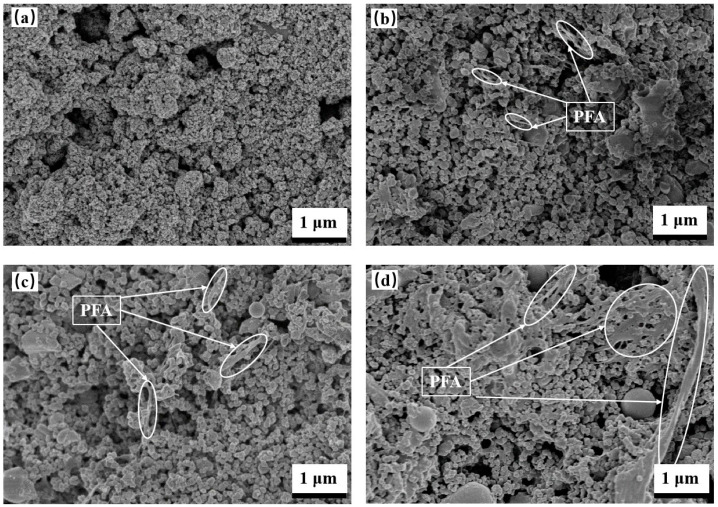
Surface high-rate SEM of coatings with different doping ratios: (**a**) 0PT, (**b**) 5PT, (**c**) 15PT, and (**d**) 25PT.

**Figure 3 nanomaterials-13-03123-f003:**
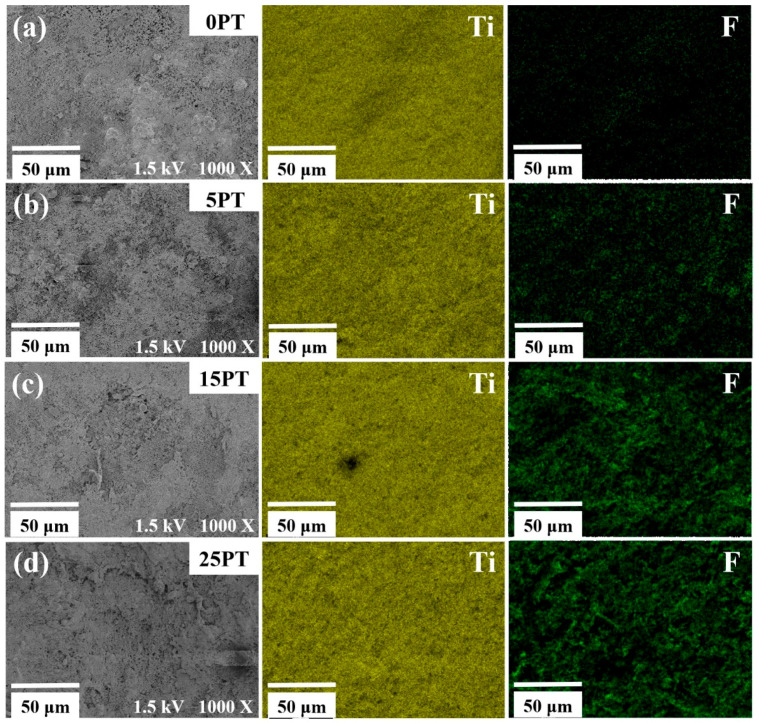
Distribution of elements on the surface of the coating under different PFA content: (**a**) 0PT, (**b**) 5PT, (**c**) 15PT, and (**d**) 25PT.

**Figure 4 nanomaterials-13-03123-f004:**
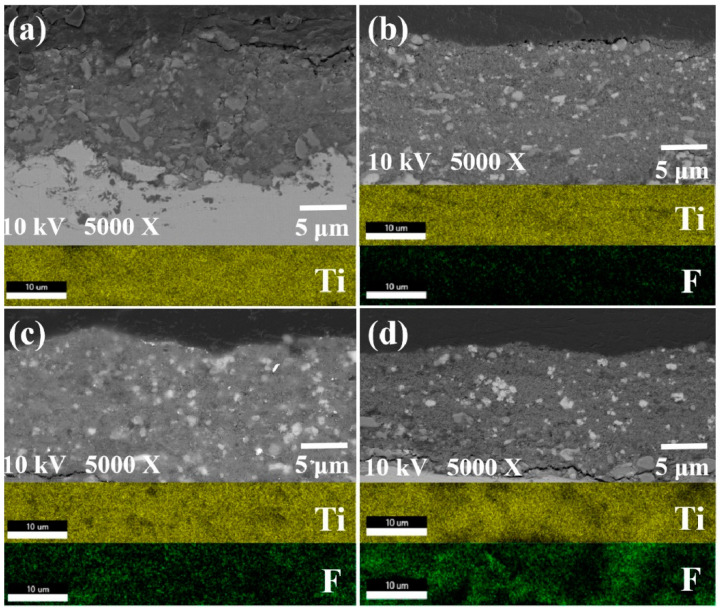
Element distribution of coating section under different PFA content: (**a**) 0PT, (**b**) 5PT, (**c**) 15PT, and (**d**) 25PT.

**Figure 5 nanomaterials-13-03123-f005:**
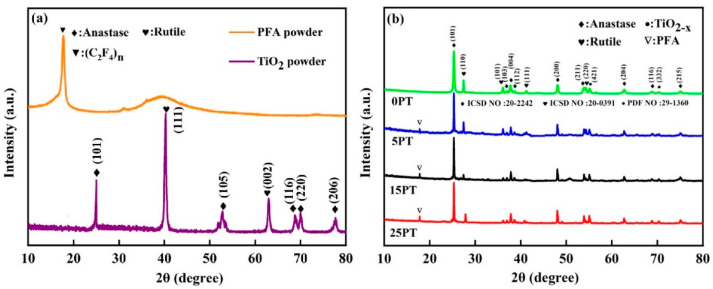
(**a**) XRD spectra of PFA and TiO_2_ powders; (**b**) XRD spectra of coatings with different PFA content.

**Figure 6 nanomaterials-13-03123-f006:**
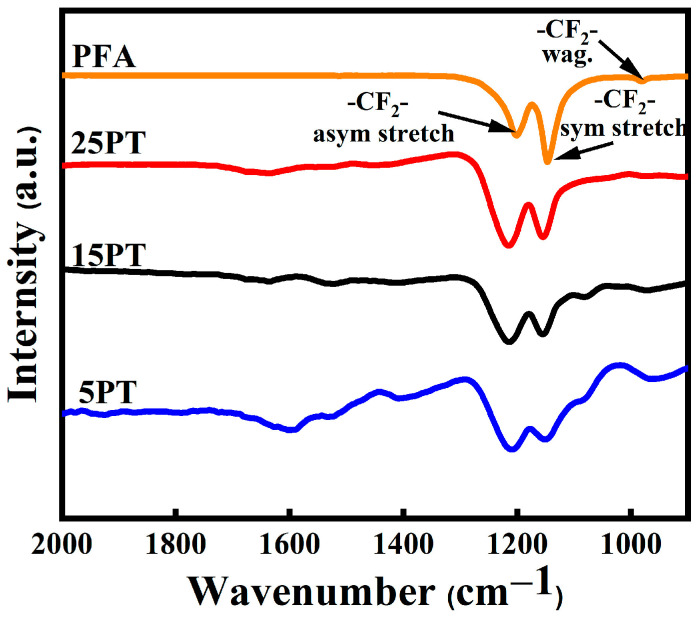
Fourier transform infrared spectrum of the sample.

**Figure 7 nanomaterials-13-03123-f007:**
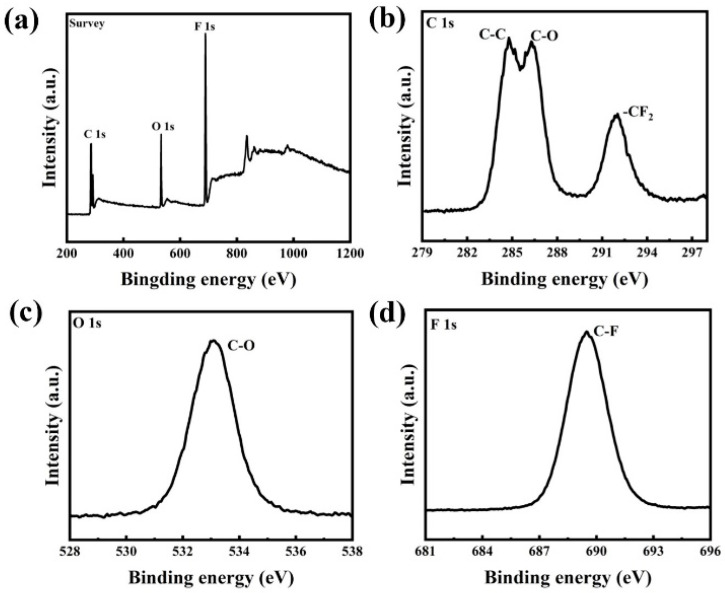
High-resolution XPS spectra of PFA powder: (**a**) whole spectra, (**b**) C 1 s, (**c**) O 1 s, and (**d**) F 1 s.

**Figure 8 nanomaterials-13-03123-f008:**
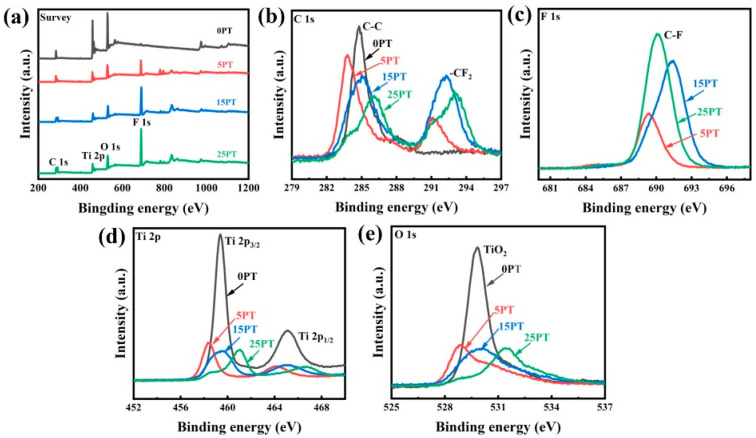
High-resolution XPS spectra of composite coatings: (**a**) whole spectra, (**b**) C 1 s, (**c**) F 1 s, (**d**) Ti 2 p, and (**e**) O 1 s.

**Figure 9 nanomaterials-13-03123-f009:**
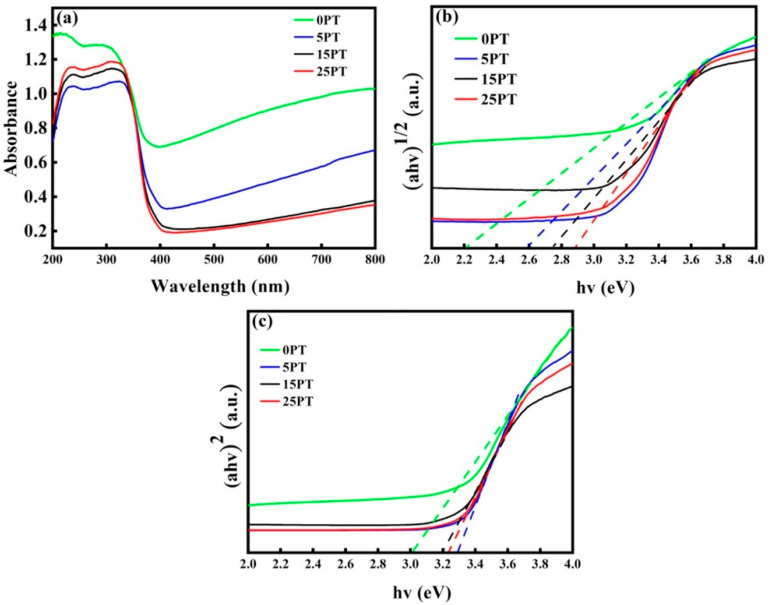
(**a**) UV–visible light absorption intensity of sample, (**b**) indirect bandgap of the sample, (**c**) direct bandgap of the sample. (The dotted line in the graph is extrapolated to the horizontal axis through the mapping method, and the intersection point is the TiO_2_ band gap value).

**Figure 10 nanomaterials-13-03123-f010:**
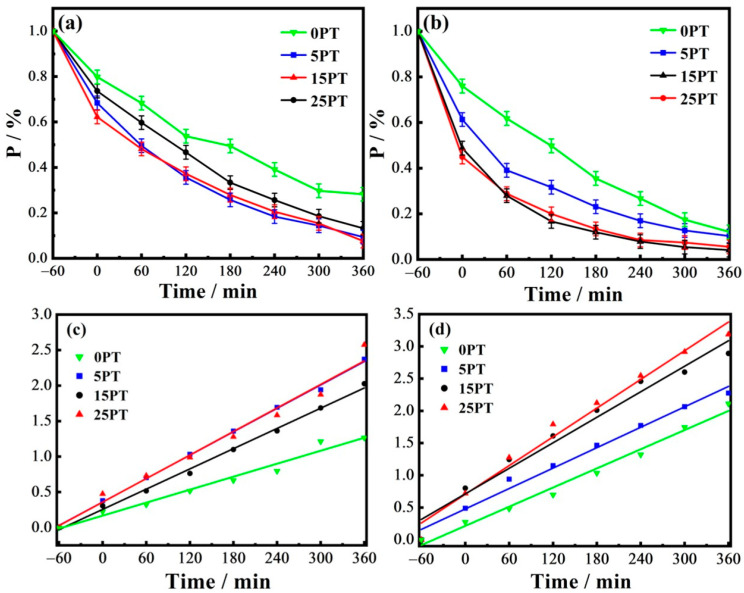
(**a**) Residual rate curve of the sample under ultraviolet light, (**b**) residual rate curve of the sample under visible light, (**c**) fitting curve of photocatalytic kinetics of the sample under ultraviolet light, (**d**) fitting curve of the photocatalytic kinetics under visible light of the sample.

**Figure 11 nanomaterials-13-03123-f011:**
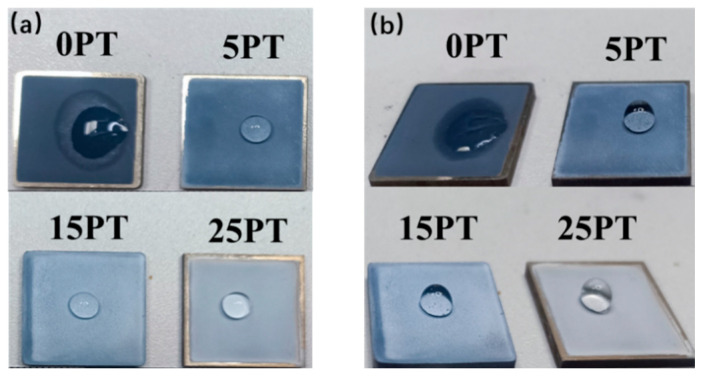
Top view (**a**), and side view (**b**), of the hydrophobic effect of the sample.

**Figure 12 nanomaterials-13-03123-f012:**
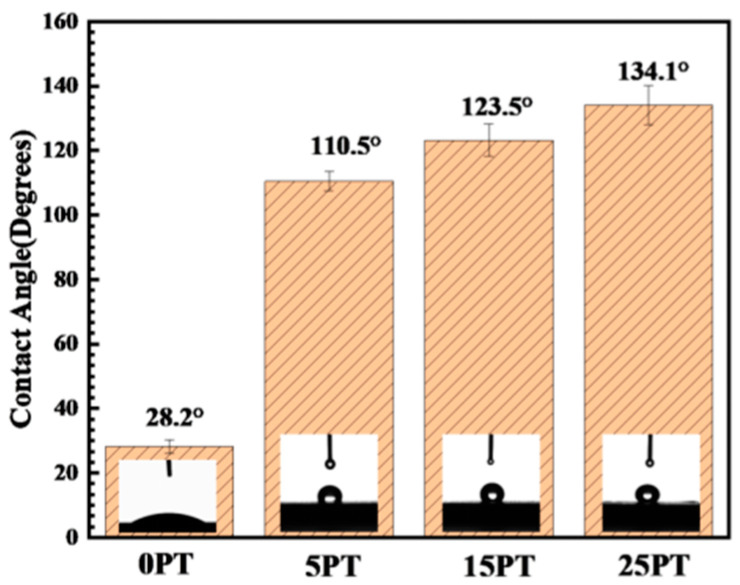
Test value of the sample contact angle.

**Table 1 nanomaterials-13-03123-t001:** Contents of components in TiO_2_-PFA suspension.

	Total Mass (g)	Solid (wt.%)	Deionized Water (g)	NNO (g)	TiO_2_ (g)	PFA Aqueous Dispersion (g)
0PT	500	15	425	1.5	75	0
5PT	500	15	416	1.425	71.25	7.5
15PT	500	15	417	1.27	63.5	18.75
25PT	500	15	386	1.125	56.25	37.5

**Table 2 nanomaterials-13-03123-t002:** Suspension plasma spraying parameters.

Simple	Coatings
Ar flow/slpm	45
H_2_ flow/slpm	7
Power/kW	27
Current/A	370
Spraying distance/mm	50

**Table 3 nanomaterials-13-03123-t003:** Arithmetic mean surface roughness (Ra) of coatings with different PFA contents.

Coating	0PT	5PT	15PT	25PT
Surface roughness parameter Ra (μm)	1.542 ± 0.5	2.418 ± 0.2	2.725 ± 0.3	4.966 ± 0.4

**Table 4 nanomaterials-13-03123-t004:** The phase content of the coating, derived from Figure 5.

Sample	0PT	5PT	15PT	25PT
Phase content anatase (%)	70	69	81	85

**Table 5 nanomaterials-13-03123-t005:** Optical direct bandgaps and indirect band gaps derived from (αhν)^2^-hν and (αhν)^1/2^-hν, respectively.

Bandgap	0PT	5PT	15PT	25PT
Indirect (eV)	2.20	2.60	2.75	2.90
Direct (eV)	3.0	3.30	3.25	3.20

**Table 6 nanomaterials-13-03123-t006:** Photocatalytic degradation of the MB, rate constants, and correlation coefficients of the fitted curves of the photocatalytic kinetics.

Simples	Light	MB Degradation (%)	Rate Constant (min^−1^)	Catalytic Rate Constant (R^2^)
0PT	UV	72	0.00305	0.97
	Vis	88	0.00496	0.98
5PT	UV	91	0.00552	0.998
	Vis	90	0.00528	0.98
15PT	UV	87	0.00475	0.996
	Vis	96	0.00661	0.96
25PT	UV	93	0.00553	0.995
	Vis	95	0.00743	0.98

## Data Availability

Suggested Data Availability Statements are available in section “MDPI Research Data Policies” at https://www.mdpi.com/ethics.

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
