# Peer review of "Novel Effective Photocatalytic Self-Cleaning Coatings: TiO2-Polyfluoroalkoxy Coatings Prepared by Suspension Plasma Spraying"

_nanomaterials, 2023, doi:10.3390/nano13243123_

Round 1
Reviewer 1 Report
Comments and Suggestions for Authors
In this manuscript the authors reported a study related to photocatalytic self-cleaning coatings based TiO2-PFA coatings prepared by suspension plasma spraying. The manuscript was written and organized well by authors. The authors conducted several analyses along with application to prove the prepared materials photocatalytic activity.
My comments are below:
1. Please add the error bar in the photocatalytic curves.
2. Please try to avoid using PFA abbreviation in title. Instated use the full name.
3. Please identify and all XRD peaks orientation in figure and discuss the results deeply in the text.
4. The introduction needs more improvement. More recent references related to the same topic need to be summarized.
5. In FTIR, the type of stretch and chemical bonding need to be identified and discussion must added.
6. Please add XPS high resolution images. Survey scan doesn’t provide enough information for the chemical state and valence information.
7. The references formatting are wrong. Please correct all of them.
Comments on the Quality of English LanguageModerate editing of English language required
Author Response
- Summary
Thanks for your professional advice and the reviewers’ instructive comments on our manuscript entitled “Novel effective photocatalytic self-cleaning coatings: TiO2-PFA coatings prepared by suspension plasma spraying”. We have revised the manuscript in accord with the comments given by you and the reviewers. We hope that the revisions can make the questions raised by the editor and reviewers clear.
- Point-by-point response to Comments and Suggestions for Authors
Reviewer 1:
In this manuscript the authors reported a study related to photocatalytic self-cleaning coatings based TiO2-PFA coatings prepared by suspension plasma spraying. The manuscript was written and organized well by authors. The authors conducted several analyses along with application to prove the prepared materials photocatalytic activity.
Response: Thank you for your careful review and constructive comments on the manuscript. We have carefully checked the whole paper and corrected the mistakes as well as the improper expression and structure. The revisions were marked in red in the revised manuscript.
Comments 1: Please add the error bar in the photocatalytic curves.
Response 1: Thank you for your suggestion. In the revised manuscript, we have added the error bar in the photocatalytic curves.
Comments 2: Please try to avoid using PFA abbreviation in title. Instated use the full name.
Response 2: Thank you for your suggestion. In the revised manuscript, we have changed the title to Novel effective photocatalytic self-cleaning coatings: TiO2- Polyfluoroalkoxy coatings prepared by suspension plasma spraying.
Comments 3: Please identify and all XRD peaks orientation in figure and discuss the results deeply in the text.
Response 3: Thank you for your question and suggestion. We have labeled the orientation of all XRD peaks in the figure and deeply discussion of the results was marked in red in the revised manuscript.
Comments 4: The introduction needs more improvement. More recent references related to the same topic need to be summarized.
Response 4: Thank you for your suggestion. In the revised manuscript, we have further improved the introduction and summarized more recent relevant references.
Comments 5: In FTIR, the type of stretch and chemical bonding need to be identified and discussion must added.
Response 5: Thank you for your question and suggestion. We have identified the type of stretch and chemical bonding in the figure and deeply discussion of the results was marked in red in the revised manuscript.
Comments 6: Please add XPS high resolution images. Survey scan doesn’t provide enough information for the chemical state and valence information.
Response 6: Thank you for your suggestion. In the revised manuscript, we have added XPS high resolution images.
Comments 7: The references formatting are wrong. Please correct all of them.
Response 7: We apologize for this mistake. In the revised manuscript, we have corrected all references.
Reviewer 2 Report
Comments and Suggestions for Authors
The authors report the preparation of TiO2-polyfluoroalkoxy (PFA) coatings and their use as photocatalysts for the degradation of the methylene blue (MB) dye. The characterization of the coatings prepared by suspension plasma spraying is rather complete but the photocatalytic part of the manuscript must markedly be improved before acceptance. Results must also be better discussed in the context of literature. The manuscript contains also many typing errors and should be carefully checked by the authors. Here are my comments :
- figure 2 : if possible SEM images with a higher magnification should be provided and size distributions added.
- Can the authors comment on the porosity of the TiO2/PVA films produced and it may markedly influence the photocatalytic activity ?
- PVA is not "doped" into the films (the term doping is not appropriate). The authors prepared composite films associating TiO2 and PVA.
- figure 3 : the distribution of the F element can only hardly been seen. Improve the mapping or the quality of the images.
- paragraph 3.2 : results must be discussed in the context of literature (influence of the PVA amount on the cristallinity of TiO2).
- the reference section contains many typing errors.
- figure 6 : correct the legend of the y axis (intensity (a.u.)) and assign all signals in the FT-IR spectra.
- Figure 7 and the related text : results must be discussed in the context of literature and all XPS signals assigned.
- Figure 8 : correct the y axis of figure 8a (absorbance and not intensity) and remove (a.u.) and the absorbance is without unity. As previously, discuss the results in the context of literature.
- Figure 9 : homogenize all figures, the y axis must start at 0. The x axis is not the wavelength but time.
- the authors must comment on the stability and on the reusability of the TiO2/PVA films.
- the authors must compare the performance of the TiO2/PVA films for the degradation of MB to other TiO2-based films and highlight the advances made.
Comments on the Quality of English LanguageMinor corrections are required.
Reviewer 3 Report
Comments and Suggestions for Authors
The manuscript " Novel effective photocatalytic self-cleaning coatings: TiO2-PFA coatings prepared by suspension plasma spraying, is interesting work, However, the comments required to address the below comments-major.
1. What is the role of the PFA for self-cleaning applications, the authors should clearly discuss in mechanistic part.
2. The authors add more information about PFA’s recent trend.
3. All figures Should be improving the quality of the figures significantly?
4. The authors should write an impressive conclusion, the present state conclusion inconstant
5. Figure 2 Label, which sample was measured? SEM images. The captured SEM images are not clearly visible make sure to provide clear images. From Figure 2 to Figure 4?
6. Figure 5 requires to modification X-Axis Intensity(a.u.) to Intensity (a.u.)- Required to space
Y-axis 2Ï´/ (°) to 2Ï´ (degree) and authors should add more discussion and also mission Index number and Plane refer Following papers and cite appropriate places: https://doi.org/10.1016/j.ceramint.2020.07.122,
7. Figure 6 letter mistakes and Required to many changes (1) X-Axis Intensity (α.u.) to Intensity (a.u.), (2) Y-axis Wavenumber(cm-1) to Wavenumber (cm-1) and Required more Discussion?
8. Figure 7 requires modifications from Counts/s to Counts (a.u.) Binding Energy(eV) to Binding Energy (eV) and more Discussion correlate with bonding interaction and self-cleaning applications
9. Figure 8 modifications Intensity(a.u.) to Intensity (a.u.) and Y-axis Wavelength/nm to Wavelength (nm) and another figure carefully checks the spacing
10. Figure 9 modifications Intensity(a.u.) to Intensity (a.u.) and Y-axis Wavelength/nm to Wavelength (nm) and another figure carefully checks the spacing
11 Figure 11 Modification Contact angle(degrees) to Contact angle (degrees)
12. The present form is very hard to revise in this paper, the authors should carefully Double revision owning to many errors. In addition, cite the paper's appropriate places https://doi.org/10.1016/j.snb.2022.131503
Round 2
Reviewer 2 Report
Comments and Suggestions for Authors
Most of the corrections suggested by the reveiwers were conducted. The following comments should still be considered :
- the language must still be improved. See for example figure 9 and the related text. The authors provided "UV-visible absorption spectra of the samples" and not "UV-visible light absorption intensity of samples" as indicated in the text. The manuscript contains many errors of this type that hinder reading.
- some sentences were added compared to the first version of the manuscript but the text was not homogenized.
- clarify/revise the paragraph (and Table 5) related to direct and indirect bandgap of the coatings. TiO2 is an n-type semiconductor with an indirect band gap of 3.2 eV.
-
Comments on the Quality of English LanguageAs indicated in my comments to authors, the language must still be improved and numerous parts of the text clarified.
Author Response
Reviewer 2:
Comments 1: The language must still be improved. See for example figure 9 and the related text. The authors provided "UV-visible absorption spectra of the samples" and not "UV-visible light absorption intensity of samples" as indicated in the text.The manuscript contains many errors of this type that hinder reading.
Response 1: Thank you for your suggestion. In the revised manuscript, the language have be improved, and the description of figure 9 has been updated. Besides, other mistakes also have been amended in the revised manuscript
Comments 2: Some sentences were added compared to the first version of the manuscript but the text was not homogenized.
Response 2: Thank you for your suggestion. In the revised manuscript, we have optimized the language and homogenized the article.
Comments 3: Clarify/revise the paragraph (and Table 5) related to direct and indirect bandgap of the coatings. TiO2 is an n-type semiconductor with an indirect band gap of 3.2 eV.
Response 3: Thank you for your question and suggestion. The decrease of TiO2 band gap is due to the fact that in the process of coating spraying, H2 reduces TiO2 at high temperature to realize the self-doping modification of Ti3+, thereby reducing the band gap and improving the catalytic ability of the catalyst under visible light.
Reviewer 3 Report
Comments and Suggestions for Authors
The author's queries are well answered, and the Present form was accepted for publication and production
